# Sex Affects Human Premature Neonates’ Blood Metabolome According to Gestational Age, Parenteral Nutrition, and Caffeine Treatment

**DOI:** 10.3390/metabo11030158

**Published:** 2021-03-09

**Authors:** Marianna Caterino, Margherita Ruoppolo, Michele Costanzo, Lucia Albano, Daniela Crisci, Giovanni Sotgiu, Laura Saderi, Andrea Montella, Flavia Franconi, Ilaria Campesi

**Affiliations:** 1Department of Molecular Medicine and Medical Biotechnology, University of Naples ‘Federico II’, 80131 Naples, Italy; marianna.caterino@unina.it (M.C.); michele.costanzo@unina.it (M.C.); 2CEINGE—Biotecnologie Avanzate Scarl, 80145 Naples, Italy; albano@ceinge.unina.it (L.A.); crisci@ceinge.unina.it (D.C.); 3Clinical Epidemiology and Medical Statistics Unit, Department of Medical, Surgical and Experimental Sciences, University of Sassari, 07100 Sassari, Italy; gsotgiu@uniss.it (G.S.); lsaderi@uniss.it (L.S.); 4Department of Biomedical Sciences, University of Sassari, 07100 Sassari, Italy; montella@uniss.it; 5Laboratory of Sex-Gender Medicine, National Institute of Biostructures and Biosystems, 07100 Sassari, Italy; flavia.franconi@gmail.com

**Keywords:** preterm infants, sex differences, targeted metabolomics, amino acids, acylcarnitines, parenteral nutrition, caffeine

## Abstract

Prematurity is the leading cause of neonatal deaths and high economic costs; it depends on numerous biological and social factors, and is highly prevalent in males. Several factors can affect the metabolome of premature infants. Accordingly, the aim of the present study was to analyze the role played by gestational age (GA), parenteral nutrition (PN), and caffeine treatment in sex-related differences of blood metabolome of premature neonates through a MS/MS-based targeted metabolomic approach for the detection of amino acids and acylcarnitines in dried blood spots. GA affected the blood metabolome of premature neonates: male and female very premature infants (VPI) diverged in amino acids but not in acylcarnitines, whereas the opposite was observed in moderate or late preterm infants (MLPI). Moreover, an important reduction of metabolites was observed in female VPI fed with PN, suggesting that PN might not satisfy an infant’s nutritional needs. Caffeine showed the highest significant impact on metabolite levels of male MLPI. This study proves the presence of a sex-dependent metabolome in premature infants, which is affected by GA and pharmacological treatment (e.g., caffeine). Furthermore, it describes an integrated relationship among several features of physiology and health.

## 1. Introduction

Prematurity, defined as childbirth before 37 weeks of gestation, annually affects 9.6% of pregnancies [1,2], being the 84% of preterm births of the so-called moderate or late preterm infants (MLPI) [3]. Relevantly, prematurity may have negative consequences for the family and the infants, during both child and adult life, and the healthcare system. Prematurity is, in fact, the leading cause of neonatal deaths and high economic costs [4]. Prematurity depends on numerous biological and social factors, such as gender, age, and smoking [5]. Notably, its incidence is higher in male neonates [6].

Several factors, including gestational age (GA) at birth, can affect qualitatively and quantitatively the metabolites measured in premature infants [7]. It can be associated with lowered maturation of metabolic pathways, elevation of catabolic stress, and other numerous confounding variables (e.g., feeding) [7,8,9]. Preterm infants can have feeding difficulties and then weight loss and growth difficulties [3]; consequently, early parenteral nutrition (PN) is recommended. A typical PN includes carbohydrates (mostly glucose), proteins (both essential and non-essential amino acids), lipids, and vitamins [10].

Moreover, preterm infants often present respiratory symptoms (apnea, intermittent hypoxemia, and extubation in mechanically ventilated premature newborns [11]) which are treated with caffeine. This methylxanthine increases respiratory rate and minute volume, stimulates respiratory centers, and increases pulmonary blood flow and the sensitivity of central medullary areas to hypercapnia, acting as a competitive inhibitor of A1 and A2A receptors of adenosine [12]. In addition, caffeine and its metabolites (especially paraxanthine) may enhance the inhibition of systemic enzymes, which modulate lipid and glucose metabolisms [13,14]. In adults, the predominant enzyme in caffeine metabolism is CYP1A2, which is more active in men [15]; however, caffeine metabolism does not seem to present a sexual dimorphism [16].

Recently, gender and sex differences have been highlighted in healthy and ill individuals [17,18]. Gender mainly refers to the socially constructed identities of individuals, whereas sex refers to the fundamental biological disparities between males and females [15,19]. Gender and sex differences start in utero and Clarke reported sex differences in birth outcomes since 1786 [20]. Moreover, they are influenced by GA at birth [7,21]. However, the influence of sex and gender is still neglected in many studies [7,22]. Although Gucciardi and colleagues did not find sex differences in acylcarnitines in neonates after different GA [23], others found sex differences in blood amino acids and acylcarnitines and urinary organic acids of at-term neonates [24,25,26].

As a matter of fact, the metabolome of biological fluids of preterm infants collected during the first days after birth shows a specific urinary and blood pattern, with high variability in amino acids, organic acids, and other metabolites [27,28,29]. 

Thus, the aim of the present study was to assess the role played by GA in the occurrence of sex-related differences in the blood metabolome of premature neonates through a targeted metabolomic approach for the detection of amino acids and acylcarnitines in dried blood spots (DBS). In addition, it was evaluated whether the sex effect was influenced by feeding condition (PN vs no parenteral nutrition (NPN)) and by caffeine treatment.

## 2. Results

### 2.1. Populations

The population of preterm infants was stratified according to sex, birth weight (BW), GA, and treatment with caffeine or PN. As expected, both male and female very premature infants (VPI) significantly weighted less than male and female MLPI. Moreover, male MLPI fed with PN had lower BW and GA than those exposed to NPN (Table 1). Caffeine-treated MLPI showed a lower GA than those not treated (Table 1).

### 2.2. GA Effect on Blood Metabolome of Premature Female and Male Babies

#### 2.2.1. Intra-Sex Analysis of GA Effect

A total of 53 variables were measured (Appendix A) from newborns-derived DBS within a platform set to detect metabolism errors. Both female and male VPI had higher levels of C4, C5, C8, and C4OH in comparison with those of MLPI, being C4 and C5 the most different in female and male neonates, respectively (Table 2). Interestingly, sexes were associated with small specific metabolic variations: VPI females had higher Tyr, C5:1, C14:2 when compared with MLPI, whereas VPI males had lower Ala and Glu and higher C3, C3DC, and C10:2 than those measured in MLPI (Table 2). Other metabolites were not statistically different (Appendix A).

Then, partial least squares-discriminant analysis (PLS-DA) was carried out to identify metabolites closely associated with sex-dependent GA according to their variable importance on projection (VIP) score. In the female cohort, Tyr, C12, and C18:1OH differentiated between VPI and MLPI cohorts (VIP > 2.0), whereas in the male cohort, C5, C5OH, and C4 (VIP > 2.0) differentiated between VPI and MLPI (Appendix A).

#### 2.2.2. Inter-Sex Analysis of GA Effect

Female and male VPI diverged in their amino acid profile (Figure 1A), with Ala, Tyr, Asp, Glu, Gly, and Cit being significantly higher in female infants. Acylcarnitines and total esterified carnitine did not show sexual dimorphism in VPI (Appendix A). By contrast, female and male MLPI diverged in acylcarnitine levels (Figure 1B): C2, C6, C4DC, C18:1, C18:2, and total esterified carnitine were higher in males. Instead, amino acid levels did not diverge between sexes in MLPI (Appendix A).

PLS-DA analysis shows that C12 and Asp (Appendix A) were discriminant between VPI females and males. C2, C18:1, and C10DC differentiated MLPI females and males (Appendix A).

### 2.3. PN Effect on Blood Metabolome of Female and Male Premature Neonates

#### 2.3.1. Intra-Sex Analysis of PN Effect

In female VPI, PN lowered 9 metabolites (Asp, Glu, Gly, Cit, ArgSuc, C5OH, C4DC, C12:1, and C18OH) and significantly increased C18:2 of ~73% in comparison with that found in the NPN group (Table 3). In female MLPI, PN lowered only Glu levels and increased C5, C12, C6DC, C14:1, C8DC, and C10:2 (Table 3). All the other metabolites were not modified (Appendix A).

In male VPI, PN increased only Phe of ~20%, whereas C2, C8, C10:2, C14-OH, total esterified carnitine, and total esterified/C0 ratio were lowered (Table 3). In male MLPI, PN lowered Asp, Glu, and ArgSuc, and increased C5, C10:1, C10, C12, C6DC, and C14:1 (Table 3). All the other metabolites were not modified (Appendix A).

PLS-DA showed that ArgSuc, Glu, Orn, and C12 could discriminate between female VPI fed with NPN and PN (Appendix A), whereas C12 and total esterified/C0 discriminated between NPN female MLPI and PN cohorts (Appendix A). C5DC and total esterified/C0 ratio could discriminate between male VPI fed with NPN and PN (Appendix A). C12, Glu, and C14 discriminated between male MLPI fed with NPN and PN (Appendix A). 

#### 2.3.2. Inter-Sex Differences of PN Effect

NPN females had higher Gly and Cit concentrations and lower levels of C8, C16, and C18:2 when compared with NPN males (Figure 2). All the other metabolites were not modified (Appendix A). PLS-DA showed that males and females were differentiated by altered level of discriminant metabolites C18:1OH, C6:1, Val, Orn, and Tyr (Appendix A). No sex-related differences were observed in infants fed with PN (Appendix A), even if C18:1 and the total esterified/C0 ratio discriminated male and female in the PN group (Appendix A).

### 2.4. Caffeine Effect on Blood Metabolome of Female and Male Premature Neonates 

#### 2.4.1. Intra-Sex Analysis of Caffeine Effect

Female VPI treated with caffeine were different only for Glu and C5:1, whose levels were higher when compared with those of non-treated individuals (Figure 3A), whereas female caffeine-treated MLPI did not show any significant differences in comparison with non-treated cases (Appendix A).

In male VPI, caffeine significantly increased Val and Xle blood levels (Figure 3B). In MLPI males, the treatment significantly affected 8 parameters: Asp and Gly were lower in the caffeine-treated cohort, whereas the acylcarnitines C3, C5, C5OH, C14:2, C4OH, and C8:1 were significantly higher (Table 4). Differences ranged from −14.3% (Gly) to 36.4% (C5OH). All the other metabolites were not modified (Appendix A).

PLS-DA showed that VIP and MLPI caffeine-treated females were discriminated by ArgSuc, C6:1, Orn, and by C18:2, C16, respectively (Appendix A). VIP and MLPI caffeine-treated males were differentiated by C14:1, C6:1, Asp, C18:2 (Appendix A) and by C4DC when compared with those not treated, respectively (Appendix A).

#### 2.4.2. Inter-Sex Differences of Caffeine Effect

Caffeine-treated females had higher Gly and lower C6 and C18:1 in comparison with those of caffeine-treated males (Figure 3C). In caffeine-free infants, Gly and Cit were significantly higher in females (Figure 3D). All the other metabolites were not modified (Appendix A).

PLS-DA showed that C5DC and Asp could differentiate male and female babies treated with caffeine (Appendix A), whereas C10DC, C18OH, C14, and C18:1 were key sex discriminants in caffeine-free infants (Appendix A).

### 2.5. Cluster Analysis of Cohorts

Cluster analysis showed four different clusters: the first one included 57 subjects, whereas the second one included 81 individuals. The third and fourth ones grouped 63 and 109 infants, respectively (Table 5). Cluster 1 included 63.2% of males and grouped almost all subjects with higher values of acylcarnitines.

Cluster 2 included VPI infants only (43.2% were males), with the lowest values of BW, GA, and several acylcarnitines (red in Table 5). 

Cluster 3 had the largest percentage (66.7%) of males and grouped almost all subjects treated with caffeine, showing that caffeine highly affects males, in agreement with Kruskall–Wallis analysis. 

Cluster 4 included most of all late preterm infants (55.1% were males), characterized by the highest values of BW and GA and the lowest values of several acylcarnitines (red in Table 5).

### 2.6. Correlation Analysis

Correlations between BW, GA, and parameters are reported in Figure 4. In females, 17 metabolites negatively correlated with BW and 10 with GA (C10DC was positively correlated, whereas the others only negatively). In males, 12 metabolites negatively correlated with BW and 16 with GA (6 positively and 10 negatively).

Furthermore, for both sexes there was a positive correlation between BW and GA (coefficient: 0.468; *p*-value < 0.001 for males; coefficient: 0.503; *p*-value < 0.001 for females).

## 3. Discussion

The present study confirms that different GA impact qualitatively and quantitatively on blood metabolome of premature neonates with corresponding change in several amino acids, free carnitine, and acylcarnitines. However, some main novelties may be elicited: (i) the evidence of quantitative and qualitative sex differences in the blood of premature neonates; (ii) the sex differences are affected by GA, type of nutrition, and caffeine. Prematurity is characterized by sex-specific levels of metabolites: VPI females had higher Tyr, C5:1, and C14:2 when compared with MLPI ones, whereas VPI males had lower Ala and Glu, as well as higher C3, C3DC, and C10:2 than those found in MLPI. Interestingly, GA also affects sex differences, with their onset being closely related to the specific GA. Male and female VPI showed different amino acids levels with the only exception of acylcarnitines; the opposite scenario was observed in MLPI (sex difference found in acylcarnitines but not in amino acids), making infants at different stages of prematurity metabolically distinct, as confirmed by the cluster analysis. This is in agreement with what we previously reported studying term infants: females had higher amino acids concentrations, whereas males had higher acylcarnitines [24]. Previously, Wilson et al. reported that Arg, leucine, Orn, Phe, and Val levels increase as a function of prematurity [7]; however, they do not assess the effects of sex. A recent study showed that 1459 metabolites, including amino acids, carbohydrates, dipeptides, lipids, nucleotides, polyamines, and xenobiotics significantly correlate with GA [30]. However, this study did not evaluate the sex influence on the metabolome. Indeed, our study assesses metabolome changes related to GA that are also influenced by sex. Prematurity may influence analyte levels through several different mechanisms related to an increased catabolic stress, prematurity of metabolic pathways, and/or organ systems (renal and hepatic) [8,31,32].

PN induced qualitative changes in the blood metabolome when the sex of babies and the grade of prematurity was considered. Preterm infants receive PN, which leads to a reduction in the levels of nine metabolites (Asp, Glu, Gly, Cit, ArgSuc, C5OH, C4DC, C12:1, and C18OH), ranging from 8% to 52% in female VPI. The decrease is less relevant in male VPI, ranging from 12% to 20% for six metabolites (C2, C8, C10:2, C14-OH, total esterified carnitine, and total esterified/C0 ratio). The dramatic reduction of metabolites in VPI females fed with PN might suggest that PN could not address a female infant’s nutritional needs, indicating the need of sex specific PN. In female MLPI, PN caused an increase (ranged from 10% to 100%) of six metabolites (C5, C12, C6DC, C14:1, C8DC, and C10:2) and a decrease of Glu. In male MLPI, a significant increase (ranged from 17% to 50%) was observed for six metabolites (C5, C10:1, C10, C12, C6DC, and C14:1), as well as a decrease of Asp, Glu, and ArgSuc. In Asian preterm infants, it has been reported that leucine, Met, tryptophan, histidine, Arg, Glu, Asp, and serine significantly increase after PN, while Phe, Tyr, Gly, and Ala decrease. However, this study did not consider the sex [33]. In addition, comparing VPI versus MLPI it emerges that the metabolism of amino acids and acylcarnitines changes with GA, being that these modifications sex-dependent. 

PN administration can lead to serious complications, being inappropriate or unsuitable for infants’ metabolic demand [34,35,36], so that a balance between the demand and intake of nutrients appears critical. As an example, earlier and higher intravenous amino acid and lipid intakes particularly increased the risk of metabolic acidosis in VPI, depending also on GA [37]. The results of our study indicate that sex is another variable that must be considered, highlighting the need to evaluate PN according to GA and sex.

The sex-specificity of the effect exerted by caffeine on the metabolome is an innovative result, at least in prematures. Caffeine causes relevant metabolic effects, modulating lipid and glucose metabolism [13,14]. In particular, Altmaier observed a negative association between caffeine and plasma concentrations of long- and medium-chain acylcarnitines in adults [38]. Caffeine decreases the levels of branched-chain and aromatic amino acids in plasma of male rats or of human adults, whereas it has no significant effects on large neutral amino acids concentrations [39,40]. It is not clear if the metabolic actions of caffeine are affected by sex; however, some adenosine activities are sex-dependent [41,42], and it has been reported that caffeine metabolism is more active in female than male preterm neonates [43]. Nevertheless, the major metabolizing enzyme of caffeine, CYP1A2, is more active in adult men than in women [15]. Caffeine activity showed the highest impact on male MLPI metabolites levels, leading to a reduction in Asp and Gly levels, and increasing C3, C5, C5OH, C14:2, C4OH, and C8:1, and favoring sex differences in Gly, C6, and C18:1. This aspect was also confirmed by cluster analysis, where almost all subjects treated with caffeine were included in Cluster 3, which also contained the largest percentage of males. The optimal dose of caffeine should be carefully chosen in agreement with GA and sex when treating preterm infants to avoid potential metabolism alterations, which could affect growth trajectories and the risk of disease.

The current study presents some limitations associated with the lack of specific information on maternal nutrition and specific data on mothers (age, health conditions, etc.), including the reason of premature birth and the mode of birth (cesarean or spontaneous delivery). Furthermore, PN received by preterm infants, even if based on pediatric protocols, was not identical because PN formulations were supplied by different pharmaceutical companies.

## 4. Methods

### 4.1. Populations

The recruited sample included 311 preterm infants (137 females and 174 males). It was divided into 2 groups according to the WHO [43] GA classification, where preterm birth is any birth before 37 completed weeks of gestation, or fewer than 259 days since the first day of the woman’s last menstrual period; VPI, with any birth between 28–31 completed weeks of gestation; and MLPI, with any birth between 32–36 completed weeks of gestation.

Preterm infants were characterized by several pathological conditions, requiring the administration of pharmacological therapies. Due to the high variability of pharmacological treatments in preterm infants, subjects treated with dopamine and glucose solutions were excluded, as well as infants with jaundice, those exposed to blood transfusions, and those fed with carnitine solutions. Characteristics of the cohorts stratified by gender and GA are summarized in Appendix A.

DBS were collected from hospitals participating in the newborn screening program of Campania Region (Italy). All experiments were performed in compliance with national laws and institutional guidelines, approved by Italian Ministry of Health in law no. 167 (19 August 2016).

### 4.2. Sample Preparation for Tandem Mass Spectrometry

Blood samples were collected from the heel of newborns between 48 and 72 h of life on a special filter paper, dried overnight at room temperature, and processed for tandem mass spectrometry (MS/MS) analysis, as previously described [24,44,45,46]. Metabolites were extracted in 200 µL of methanol containing labelled amino acids and acylcarnitines standards, (^15^N,2-^13^C-Gly; ^2^H_4_-Ala; ^2^H_8_-Val; ^2^H_3_-Leu; ^2^H_3_-Met; ^13^C_6_-Phe; ^13^C_6_-Tyr; ^2^H_3_-Asp; ^2^H_3_-Glu; ^2^H_2_-Orn; ^2^H_2_-Cit; ^2^H_4_,5-^13^C-Arg; and ^2^H_9_-C0; ^2^H_3_-C2; ^2^H_3_-C3; ^2^H_3_-C4; ^2^H_9_-C5; ^2^H_3_-C8; ^2^H_9_-C14; ^2^H_3_-C16) at concentrations of 500–2500 µM and 7.6–152 µM, respectively. Samples were incubated at room temperature for 20 min and dried under nitrogen flow. Metabolites were derivatized using 80 µL n-butanol/3N HCl at 65 °C for 25 min. Samples were dried and resuspended in 300 µL of acetonitrile/water (70:30) containing 0.05% formic acid. A final volume of 40 µL was used for MS analysis.

### 4.3. Tandem Mass Spectrometry Analysis 

Metabolites were identified and quantified by MS/MS using an API 4000 triple quadrupole mass spectrometer (Applied Biosystems-Sciex, Toronto, Canada) coupled with a high performance liquid chromatograph (1200 series; Agilent Technologies, Waldbronn, Germany).

Acylcarnitines and amino acid quantification was performed by precursor ion scan and neutral loss scan, respectively; glycine (Gly), ornithine (Orn), arginine (Arg), and citrulline (Cit) were quantified by multiple reaction monitoring (MRM). Precursor ion scan was performed using the following parameters: precursor ion mass: 85.1 Da; polarity: positive; *m*/*z* range (Da): 200–560; declustering potential (DP) range (volts): 55–80; collision energy (CE) range (volts): 34–60. Neutral loss scan was performed according to: neutral loss mass: 102 Da; polarity: positive; *m*/*z* range (Da): 130–280; DP (volts): 45; CE (volts): 25. Finally, MRM mode parameters were: positive polarity; Q1/Q3 (*m*/*z*): 132.1/76.0 (Gly), 189.1/70.0 (Orn), 231.2/70.0 (Arg), 232.2/113.1 (Cit); DP (volts): 43 (Gly), 33 (Orn), 60 (Arg), 50 (Cit); CE (volts): 14 (Gly), 33 (Orn), 45 (Arg), 28 (Cit). 

Quantitative analysis of the data was performed with ChemoView v1.2 software (SCIEX, Framingham, MA, USA) using stable isotope-labeled internal standards to improve matrix correction and compare analyte areas. Quality controls, used to test the mass spectrometry methods’ accuracy and precision, were provided by CDC (Centers for Disease Control and Prevention; Atlanta, GA, USA) and ERNDIM (European Research Network for evaluation and improvement of screening, Diagnosis, and treatment of Inherited disorders of Metabolism; Manchester, UK; www.erndimqa.nl).

### 4.4. Statistical Analysis 

Concentrations are reported as µM, this unit being commonly used by laboratories that perform newborn screening. Qualitative variables were described with absolute and relative frequencies, whereas quantitative variables with means (standard deviations) or medians (interquartile ranges), depending on their normal or non-normal distribution, respectively. Qualitative variables were compared with chi-squared or Fisher exact test, when appropriate. Normal and non-normal quantitative variables were compared with Student *t* or Mann–Whitney test, respectively. A Spearman correlation was performed. A factor loading of 0.4 was considered to select the variables for the cluster analysis. Visual assessment of the distribution of the clusters was based on a dendrogram. The Gower dissimilarity value was adopted for the identification of the clusters. A *p*-value lower than 0.05 was considered statistically significant. 

All statistical computations were performed with the statistical software STATA version 16 (StatsCorp, College Station, Texas, USA). Multivariate statistical analysis was performed using MetaboAnalyst 4.0 (http://www.metaboanalyst.ca) [47,48]. The dataset was processed to estimate missing values, removing the features with >50% missing values and replacing the remaining ones by using 1/5 of the minimum value of each variable. Data were log (2)-transformed and Pareto scaled. 

Partial least squares-discriminant analysis (PLS-DA) was used, and the corresponding variable importance on projection (VIP) was estimated for each differentially abundant metabolite [47].

## 5. Conclusions

This study confirms that metabolomics analysis is a powerful tool in the early characterization of the sexual specificity of perinatal, pediatric, and adulthood status. In preterm infants, the levels of amino acids and acylcarnitines depend on prematurity, PN and caffeine treatment, and on sex. Sex, in fact, induces qualitative changes in all examined conditions (GA, PN, caffeine treatment). Therefore, sex should be an independent variable in metabolomic studies. Finally, the study shows the importance of intersectionality, considering integrated relationships among the studied parameters.

## Figures and Tables

**Figure 1 metabolites-11-00158-f001:**
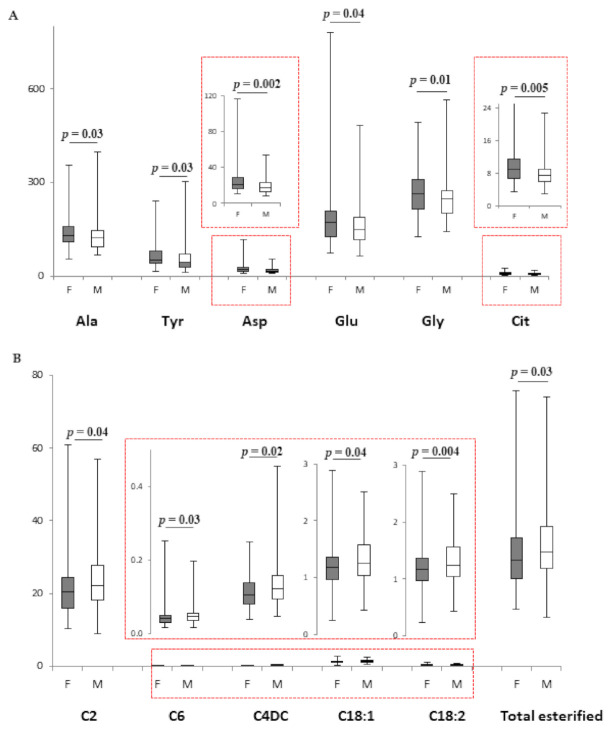
(**A**). Box plot of amino acids (µM) that displays the sex difference in female (*n* = 66; grey bars) and male (*n* = 81; white bars) VPI. (**B**). Box plot of acylcarnitines (µM) that displays the sex difference in female (*n* = 71; grey bars) and male (*n* = 93; white bars) MLPI. The horizontal line across the box represents the median, and the box comprises the first and the third quartiles. The vertical lines represent the minimum and the maximum values. Red boxes represent an enlargement of the box plots necessary due to the different scale on the *y* axis.

**Figure 2 metabolites-11-00158-f002:**
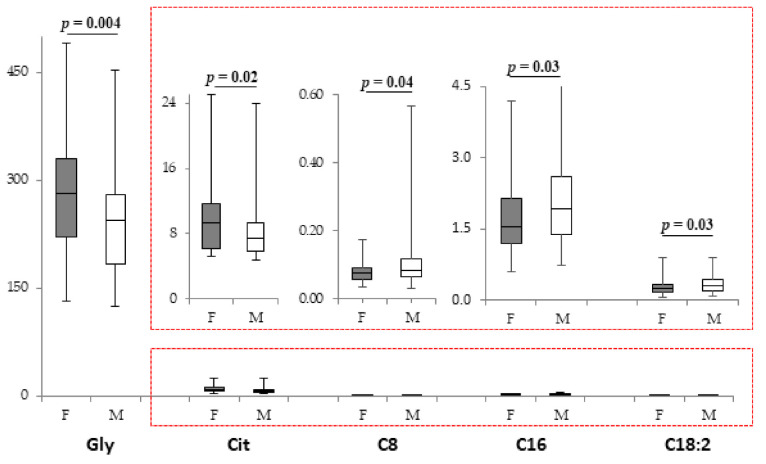
Box plot of metabolites (µM) that displays the sex difference in female (*n* = 62; grey bars) and male (*n* = 79; white bars) infants not fed with PN. The horizontal line across the box represents the median, and the box comprises the first and the third quartiles. The vertical lines represent the minimum and the maximum values. Red boxes represent an enlargement of the box plots necessary due to the different scale on the y axis.

**Figure 3 metabolites-11-00158-f003:**
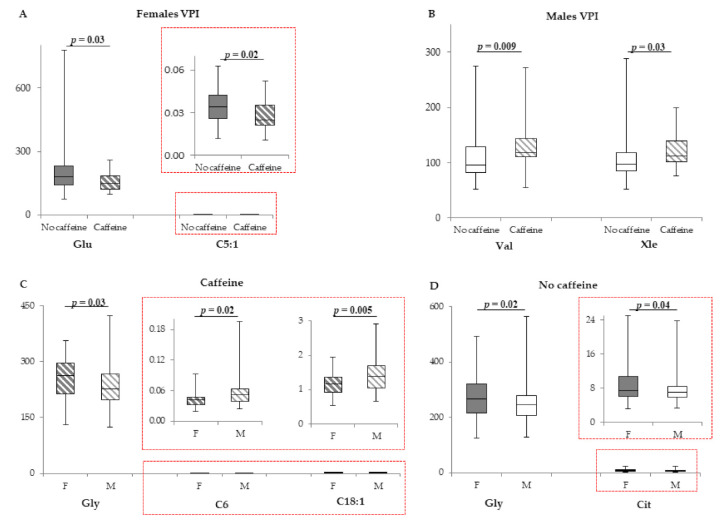
(**A**). Box plot of metabolites (µM) that displays the difference in VPI females not treated with caffeine (*n* = 41; grey bars) and treated with caffeine (*n* = 25; stripped bars). (**B**)**.** Box plot of amino acids (µM) that displays the sex difference in VPI males not treated with caffeine (*n* = 52; white bars) and treated with caffeine (*n* = 29; stripped bars). (**C**). Box plot of metabolites (µM) that displays the sex difference in females (*n* = 41; grey stripped bars) and males (*n* = 50; white stripped bars) treated with caffeine. (**D**). Box plot of metabolites (µM) that displays the sex difference in females (*n* = 96; grey bars) and males (*n* = 25; white bars) not treated with caffeine. The horizontal line across the box represents the median, and the box comprises the first and the third quartiles. The vertical lines represent the minimum and the maximum values. Red boxes represent an enlargement of the box plots necessary due to the different scale on the y axis.

**Figure 4 metabolites-11-00158-f004:**
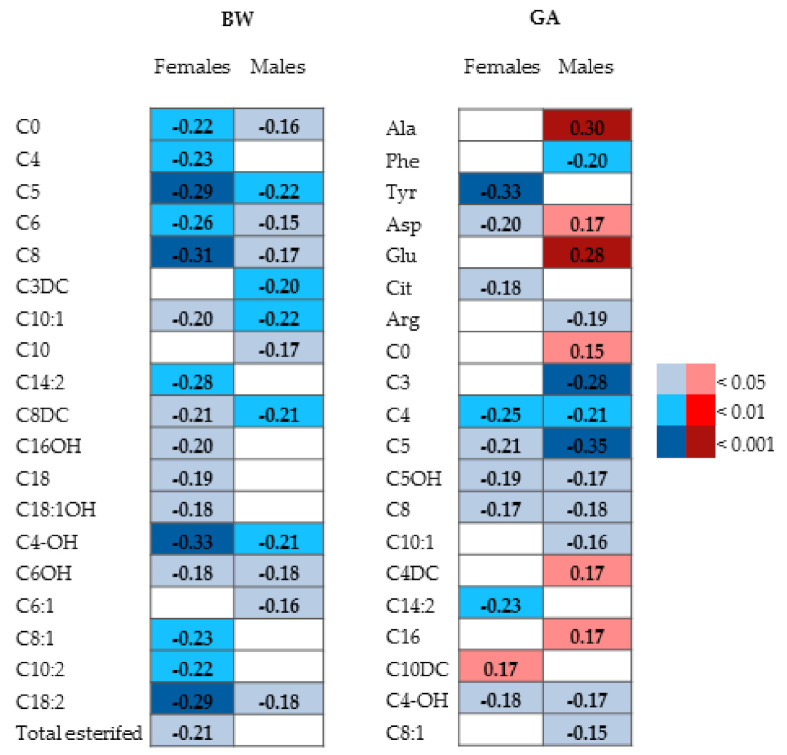
Heat maps of correlations among BW or GA in female and male neonates. Values represent the Spearman’s correlation coefficients. The red and blue colors of the cells indicate positive and negative statistically significant correlation, respectively.

**Table 1 metabolites-11-00158-t001:** Population stratification according to sex, gestational age (GA) (weeks), body weight (BW) (g) and caffeine treatment.

	Preterm Infants (311)							
	Females (137)				Males (174)			
	VPI (66)		MLPI (71)		VPI (81)		MLPI (93)	
BW	1313 (1120–1460)		1600 (1470–1700) *		1350 (1180–1470)		1545 (1430–1690) *	
GA	30 (29–31)		34 (33–34) *		30 (29–31)		33 (32–34) *	
	No caffeine (41)	Caffeine (25)	No caffeine (55)	Caffeine (16)	No caffeine (52)	Caffeine (29)	No caffeine (72)	Caffeine (21)
BW	1250	1400	1600	1539	1350	1330	1560	1500
	(1110–1400)	(1140–1580)	(1500–1700)	(1282–1703)	(1195–1452)	(1160–1564)	(1440–1683)	(1420–1700)
GA	30 (29–31)	30 (29–31)	34 (33–35)	33 (32–34) °	30 (29–31)	30 (30–31)	34 (32–35)	32 (32–33) °
	NPN (28)	PN (38)	NPN (34)	PN (37)	NPN (30)	PN (51)	NPN (49)	PN (44)
BW	1340	1210	1670	1570	1350	1330	1590	1490
	(1420–1700)	(996–1490)	(1500–1720)	(1450–1660)	(1240–1453)	(1160–1490)	(1450–1700)	(1380–1660) ^
GA	30.5 (29–31.5)	31 (29–31)	34 (33–35)	33 (33–34)	30.5 (30–31)	30 (29–31)	34 (32–36)	33 (32–34) ^

In rounded brackets after each heading, the number of samples; * very premature infants (VPI) vs. moderate or late preterm infants (MLPI) (*p* < 0.001); ° MLPI: no caffeine vs. caffeine (*p* = 0.009); ^ MLPI males: no parenteral nutrition (NPN) vs. parenteral nutrition (PN) (*p* = 0.03 for BW and *p* = 0.04 for GA).

**Table 2 metabolites-11-00158-t002:** Intra-sex analysis of GA effect in female and male preterm neonates.

**Females** **(137** **)**				
	**VPI** **(66** **)**	**MLPI** **(71** **)**	***p***	**% of Change**
Tyr	52.0 (42.9–80.8)	42.4 (30.4–60.8)	0.002	−18.5
C4	0.37 (0.26–0.50)	0.26 (0.22–0.39)	0.004	−29.7
C5:1	0.03 (0.02–0.04)	0.03 (0.02–0.03)	0.02	0.0
C5	0.23 (0.19–0.32)	0.19 (0.14–0.30)	0.03	−17.4
C8	0.08 (0.06–0.12)	0.07 (0.05–0.10)	0.02	−12.5
C14:2	0.06 (0.05–0.07)	0.05 (0.04–0.06)	0.007	−16.7
C4-OH	0.08 (0.06–0.11)	0.07 (0.06–0.10)	0.03	−12.5
**Males** **(174** **)**				
	**VPI** **(81** **)**	**MLPI** **(93** **)**	***p***	**% of Change**
Ala	124.4 (94.8–147.2)	143.2 (110.7–171.0)	0.0008	15.1
Glu	149.4 (117.5–190.2)	165.1 (136.6–215.0)	0.01	10.5
C3	3.0 (2.1–3.6)	2.4 (1.6–3.1)	0.002	−20.0
C4	0.36 (0.28–0.58)	0.30 (0.24–0.41)	0.006	−16.7
C5	0.25 (0.20–0.32)	0.20 (0.14)	<0.001	−20.0
C8	0.09 (0.07–0.13)	0.08 (0.06–0.11)	0.04	−11.1
C3DC	0.04 (0.03–0.05)	0.03 (0.03–0.04)	0.04	−25.0
C4-OH	0.09 (0.07–0.11)	0.08 (0.06–0.10)	0.01	−11.1
C10:2	0.04 (0.03–0.05)	0.03 (0.03–0.04)	0.02	−25.0

Values (median values and IQR) are expressed as µM; in rounded brackets after each heading, the number of samples.

**Table 3 metabolites-11-00158-t003:** Intra-sex analysis of PN effect in females and males.

**Females VPI** **(66** **)**	**NPN** **(28** **)**	**PN** **(38** **)**	***p***	**% of Change**
ASP	26.2 (20.0–40.7)	19.4 (14.9–24.6)	0.002	−26.0
Glu	206.6 (172.7–256.8)	150.5 (123.8–176.0)	0.0003	−27.2
Gly	283.2 (240.4–332.3)	260 (201.2–285.5)	0.01	−8.2
Cit	10.5 (8.8–12.6)	7.0 (6.3–9.7)	0.01	−33.3
C5OH	0.14 (0.12–0.17)	0.11 (0.09–0.15)	0.02	−21.4
C4DC	0.12 (0.10–0.16)	0.10 (0.08–0.14)	0.007	−16.7
C12:1	0.04 (0.03–0.05)	0.03 (0.02–0.04)	0.02	−25.0
C18OH	0.02 (0.01–0.02)	0.01 (0.01–0.02)	0.04	−50.0
C18:2	0.22 (0.17–0.37)	0.38 (0.23–0.52)	0.006	72.7
ArgSuc	0.21 (0.13–0.34)	0.10 (0.08–0.13)	<0.0001	−52.4
**Females MLPI** **(71** **)**	**NPN** **(34** **)**	**PN** **(37** **)**	***p***	**% of Change**
Glu	196.1 (142.8–261.3)	156.9 (135.7–174.3)	0.007	−20.0
C5	0.16 (0.13–0.23)	0.23 (0.17–0.33)	0.005	43.8
C12	0.09 (0.07–0.11)	0.14 (0.11–0.18)	<0.0001	55.6
C6DC	0.02 (0.02–0.03)	0.04 (0.03–0.04)	<0.0001	100.0
C14:1	0.10 (0.07–0.12)	0.11 (0.09–0.14)	0.04	10.0
C8DC	0.03 (0.02–0.03)	0.03 (0.2–0.03)	0.005	0.0
C10:2	0.03 (0.02–0.04)	0.04 (0.03–0.05)	0.02	33.3
**Males VPI** **(81** **)**	**NPN** **(30** **)**	**PN** **(51** **)**	***p***	**% of Change**
Phe	40.6 (38.7–51.2)	48.9 (40.1–54.1)	0.03	20.4
C2	22.8 (19.4–31.3)	20.0 (15.2–25.4)	0.01	−12.3
C8	0.10 (0.08–0.14)	0.08 (0.06–0.11)	0.04	−20.0
C10:2	0.05 (0.03–0.06)	0.04 (0.03–0.05)	0.02	−20.0
C14-OH	0.02 (0.02–0.04)	0.02 (0.02–0.03)	0.02	0.0
Esterified	33.0 (28.0–44.8)	28.9 (23.0–35.5)	0.03	−12.4
Esterified/C0	1.12 (0.93–1.38)	0.93 (0.80–1.03)	0.001	−17.0
**Males MLPI** **(93** **)**	**NPN** **(49** **)**	**PN** **(44** **)**	***p***	**% of Change**
Asp	19.9 (16.2–30.7)	17.4 (12.2–23.6)	0.04	−12.6
Glu	191.6 (149.9–258.2)	149.0 (124.7–189.7)	0.005	−22.2
ArgSuc	0.15 (0.10–0.27)	0.11 (0.08–0.13)	0.02	−26.7
C5	0.17 (0.13–0.26)	0.22 (0.18–0.25)	0.03	29.4
C10:1	0.06 (0.05–0.08)	0.08 (0.06–0.10)	0.02	33.3
C10	0.06 (0.04–0.08)	0.07 (0.06–0.09)	0.007	16.7
C12	0.10 (0.07–0.12)	0.15 (0.11–0.22)	<0.0001	50.0
C6DC	0.02 (0.02–0.03)	0.03 (0.02–0.04)	0.0004	50.0
C14:1	0.10 (0.08–0.12)	0.12 (0.10–0.16)	0.002	20.0

Values (median values and IQR) are expressed as µM. In rounded brackets after headings, the number of samples.

**Table 4 metabolites-11-00158-t004:** Caffeine effect in male MLPI.

Male MLPI (93)	No Caffeine (72)	Caffeine (n = 21)	*p*	% of Changes
Asp	19.9 (15.5–28.2)	15.4 (12.5–19.3)	0.01	−22.6
Gly	246.1 (209.3–308.4)	210.8 (190.1–251.3)	0.03	−14.3
C3	2.10 (1.41–3.00)	2.79 (2.22–3.74)	0.02	32.9
C5	0.20 (0.14–0.24)	0.25 (0.18–0.37)	0.03	25.0
C5OH	0.11 (0.09–0.13)	0.15 (0.13–0.17)	0.0005	36.4
C14:2	0.05 (0.04–0.06)	0.06 (0.05–0.09)	0.01	20.0
C4-OH	0.08 (0.06–0.09)	0.10 (0.07–0.11)	0.04	25.0
C8:1	0.08 (0.06–0.11)	0.10 (0.07–0.15)	0.04	25.0

Values (median values and IQR) are expressed as µM. In rounded brackets after headings, the number of samples.

**Table 5 metabolites-11-00158-t005:** Clusters analysis of cohorts.

	Cluster 1 (57)	Cluster 2 (81)	Cluster 3 (63)	Cluster 4 (109)	*p*
MLPI *n* (%)	25/57 (47.4)	0/81 (0.0)	28/63 (44.4)	108/109 (99.1)	**<0.0001**
Males, *n* (%)	36/57 (63.2)	35/81 (43.2)	42/63 (66.7)	60/109 (55.1)	**0.02**
Caffeine *n* (%)	6/57 (10.5)	17/81 (21.0)	63/63 (100.0)	4/109 (3.7)	**<0.0001**
Variables	Median (25–75°)	Median (25–75°)	Median (25–75°)	Median (25–75°)	
BW (g)	1440 (1250–1550)	1340 (1160–1455)	1400 (1215–1614)	1600 (1460–1700)	**0.0001**
GA (weeks)	31 (30–33)	30 (29–31)	31 (30–33)	34 (33–35)	**0.0001**
Ala	150.801 (122.174–180.062)	124.026 (99.062–148.067)	126.591 (107.798–151.829)	145.052 (109.841–172.662)	**0.0003**
Val	106.584 (83.963–136.743)	104.994 (86.13–127.96)	117.012 (102.017–136.597)	106.837 (85.085–126.932)	0.12
Xle	107.413 (90.137–130.46)	101.852 (88.065–117.632)	110.647 (100.696–123.947)	101.129 (83.692–128.112)	0.24
Met	17.539 (13.803–22.582)	15.973 (10.929–20.85)	16.195 (12.775–21.616)	17.51 (12.73–22.8)	0.33
Phe	47.4 (41.375–51.744)	45.185 (38.978–52.324)	44.745 (40.114–55.226)	44.545 (37.098–52.693)	0.18
Tyr	45.995 (28.449–74.207)	48.375 (36.883–70.545)	44.892 (30.827–67.913)	43.727 (33.838–68.168)	0.40
Asp	17.735 (13.168–27.255)	18.744 (13.32–24.069)	19.077 (15.92–22.421)	19.683 (16.534–27.125)	0.46
Glu	190.407 (148.687–223.336)	144.804 (115.583–181.086)	156.037 (134.021–191.755)	168.567 (135.652–211.863)	**0.0006**
Gly	276.704 (232.942–332.976)	240.244 (205.041–268.421)	252.431 (209.438–284.445)	247.257 (210.468–303.637)	**0.003**
Orn	26.499 (19.007–37.426)	27.371 (22.179–35.831)	28.965 (21.691–38.736)	26.492 (21.807–43.237)	0.66
Cit	8.213 (6.04–10.569)	7.493 (5.904–9.678)	7.421 (5.945–9.477)	7.265 (5.985–9.349)	0.72
Arg	5.952 (3.179–10.897)	6.195 (4.163–8.947)	7.518 (4.961–10.917)	5.577 (3.655–8.814)	0.24
ArgSuc	0.117 (0.086–0.163)	0.119 (0.092–0.182)	0.131 (0.099–0.216)	0.127 (0.093–0.217)	0.45
C0	47.602 (41.255–65.436)	27.103 (22.293–30.701)	32.913 (26.103–40.13)	29.325 (23.784–36.708)	**0.0001**
C2	29.484 (24.923–35.353)	18.261 (15.026–21.575)	22.15 (18.455–26.398)	20.587 (16.11–22.948)	**0.0001**
C3	3.354 (2.605–4.471)	2.275 (1.481–3.151)	2.772 (2.112–3.836)	1.977 (1.462–2.742)	**0.0001**
C4	0.465 (0.341–0.687)	0.326 (0.243–0.468)	0.368 (0.248–0.5)	0.269 (0.225–0.34)	**0.0001**
C5	0.266 (0.208–0.432)	0.223 (0.181–0.293)	0.234 (0.199–0.278)	0.183 (0.135–0.234)	**0.0001**
C6	0.06 (0.047–0.075)	0.042 (0.031–0.047)	0.047 (0.041–0.061)	0.04 (0.031–0.052)	**0.0001**
C8	0.111 (0.077–0.143)	0.076 (0.059–0.091)	0.094 (0.07–0.123)	0.07 (0.053–0.092)	**0.0001**
C10	0.103 (0.072–0.155)	0.06 (0.05–0.072)	0.065 (0.052–0.085)	0.059 (0.046–0.077)	**0.0001**
C12	0.178 (0.13–0.252)	0.092 (0.067–0.116)	0.103 (0.076–0.141)	0.106 (0.077–0.147)	**0.0001**
C14	0.307 (0.255–0.352)	0.145 (0.114–0.181)	0.179 (0.131–0.252)	0.153 (0.123–0.194)	**0.0001**
C16	3.054 (2.725–3.523)	1.226 (0.951–1.613)	1.678 (1.362–2.351)	1.549 (1.305–1.907)	**0.0001**
C18	1.119 (0.96–1.31)	0.628 (0.485–0.791)	0.826 (0.631–1.001)	0.663 (0.535–0.784)	**0.0001**
C3DC	0.043 (0.035–0.053)	0.033 (0.024–0.04)	0.037 (0.029–0.046)	0.033 (0.026–0.041)	**0.0001**
C4DC	0.134 (0.1–0.176)	0.101 (0.081–0.127)	0.119 (0.088–0.146)	0.111 (0.088–0.143)	**0.001**
C5DC	0.056 (0.04–0.074)	0.041 (0.034–0.052)	0.043 (0.035–0.058)	0.043 (0.033–0.051)	**0.0001**
C6DC	0.03 (0.024–0.04)	0.024 (0.017–0.034)	0.026 (0.017–0.035)	0.024 (0.019–0.036)	**0.005**
C8DC	0.038 (0.03–0.042)	0.026 (0.021–0.033)	0.027 (0.02–0.034)	0.026 (0.021–0.032)	**0.0001**
C10DC	0.594 (0.511–0.703)	0.256 (0.206–0.341)	0.324 (0.256–0.466)	0.313 (0.234–0.411)	**0.0001**
C4OH	0.106 (0.088–0.131)	0.074 (0.06–0.093)	0.09 (0.068–0.107)	0.068 (0.053–0.082)	**0.0001**
C5OH	0.14 (0.113–0.167)	0.116 (0.096–0.141)	0.132 (0.104–0.158)	0.104 (0.088–0.134)	**0.0001**
C6OH	0.037 (0.032–0.047)	0.032 (0.021–0.039)	0.035 (0.025–0.042)	0.029 (0.023–0.035)	**0.0001**
C12OH	0.026 (0.019–0.038)	0.018 (0.015–0.022)	0.02 (0.016–0.024)	0.019 (0.015–0.023)	**0.0001**
C14OH	0.031 (0.023–0.04)	0.02 (0.016–0.025)	0.02 (0.017–0.027)	0.021 (0.016–0.027)	**0.0001**
C16OH	0.032 (0.025–0.04)	0.02 (0.016–0.024)	0.02 (0.017–0.027)	0.018 (0.015–0.025)	**0.0001**
C16:1OH	0.061 (0.053–0.072)	0.034 (0.027–0.043)	0.041 (0.034–0.049)	0.037 (0.03–0.046)	**0.0001**
C18OH	0.017 (0.013–0.025)	0.014 (0.011–0.016)	0.013 (0.011–0.017)	0.013 (0.01–0.017)	**0.0001**
C18:1OH	0.03 (0.025–0.036)	0.021 (0.016–0.024)	0.023 (0.018–0.031)	0.02 (0.017–0.025)	**0.0001**
C5:1	0.032 (0.026–0.036)	0.03 (0.022–0.038)	0.029 (0.022–0.036)	0.028 (0.02–0.035)	0.11
C6:1	0.059 (0.044–0.086)	0.044 (0.031–0.061)	0.053 (0.041–0.068)	0.049 (0.036–0.061)	**0.002**
C8:1	0.099 (0.073–0.147)	0.086 (0.061–0.129)	0.101 (0.061–0.156)	0.075 (0.057–0.107)	**0.004**
C10:1	0.096 (0.07–0.137)	0.068 (0.046–0.092)	0.071 (0.054–0.109)	0.066 (0.051–0.086)	**0.0001**
C10:2	0.04 (0.031–0.054)	0.038 (0.028–0.047)	0.034 (0.029–0.046)	0.033 (0.025–0.042)	**0.03**
C12:1	0.056 (0.042–0.087)	0.027 (0.024–0.035)	0.033 (0.024–0.045)	0.031 (0.024–0.038)	**0.0001**
C14:1	0.166 (0.133–0.212)	0.097 (0.071–0.115)	0.115 (0.087–0.144)	0.096 (0.074–0.12)	**0.0001**
C14:2	0.064 (0.05–0.084)	0.052 (0.041–0.064)	0.057 (0.046–0.075)	0.047 (0.038–0.058)	**0.0001**
C16:1	0.257 (0.191–0.312)	0.11 (0.083–0.131)	0.118 (0.089–0.2)	0.108 (0.088–0.141)	**0.0001**
C18:1	1.72 (1.499–2.009)	0.964 (0.769–1.304)	1.326 (1.075–1.52)	1.14 (0.958–1.316)	**0.0001**
C18:2	0.339 (0.248–0.521)	0.296 (0.198–0.452)	0.32 (0.216–0.498)	0.29 (0.186–0.376)	**0.03**
Total esterified	46.269 (38.999–80.999)	26.523 (22.029–29.359)	52.47 (35.969–71.096)	36.111 (29.416–51.438)	**0.0001**
Esterified/free	0.878 (0.512–1.007)	0.987 (0.815–1.157)	0.093 (0.046–0.819)	0.71 (0.047–0.989)	**0.0001**

Values (median values and IQR) are expressed as µM. In rounded brackets after headings, the number of samples.

## Data Availability

The data presented in this study are available on request from the corresponding authors. The data are not publicly available due to privacy policy.

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
