# Peer review of "Sex Affects Human Premature Neonates’ Blood Metabolome According to Gestational Age, Parenteral Nutrition, and Caffeine Treatment"

_metabolites, 2021, doi:10.3390/metabo11030158_

Round 1
Reviewer 1 Report
The paper is well designed and the results are presented in a proper way. The methods used are adequate for this type of investigation. The discussion is mature and supported by relevant literature.
Minor remarks:
- the determination of abbreviation of measured metabolites is presented in Table S2, which reference is on page 13. The first abbreviations of those metabolites are listed on page 3 without any explenation.
- the conclusion is too meager. It should be corrected and more detailed should be added
Author Response
The paper is well designed and the results are presented in a proper way. The methods used are adequate for this type of investigation. The discussion is mature and supported by relevant literature.
We thank the referee for understanding the meaning of our work and appreciating it.
Minor remarks:
- the determination of abbreviation of measured metabolites is presented in Table S2, which reference is on page 13. The first abbreviations of those metabolites are listed on page 3 without any explanation. Table S2 (now table S1) has been moved in the results to appear earlier. Consequently, all the other supplementary Tables have been renumbered accordingly.
- the conclusion is too meager. It should be corrected and more detailed should be added. Conclusion has been deepened, as requested.
Reviewer 2 Report
The article is an interesting research, contains a large amount of data. However, the sequence of presentation and incomplete presentation of data complicates perception. So, for example, table 2 shows different indicators for male and female children, apparently in order to reduce the amount of data. In general, it seems to me that it would be more appropriate to use multidimensional methods of data processing in order to more clearly present information. The conclusions are too general from such a volume of information. I believe that the article needs revision in terms of presentation of the material and discussion of the results. There are typos in the article.
However, the authors of the article, in my opinion, chose an inappropriate way of presenting data, overloading the article with tables and figures.
1. Table 2 - only the differing values ​​for males and females are given, although it is interesting to understand the order of the differences. I would like to see the same list of parameters for both male and female.
2. Figure 1 - for children with VPI data are given for amino acids, for children with MLPI for acylcarnitines.
3. Table 3 - again a different list of parameters for each subgroup. I would recommend using a more visual way of presenting the data, for example, the PCA method. It will allow not only to reduce the dimension, i.e. reduce the number of important differing parameters, but also show their relationship.
4. Very difficult to read table 5. How are p values ​​in the table calculated? Was the Bonferroni Amendment used? In this case, the Kruskal-Wallis test can be used, since the distribution of the data is most likely not normal.
5. The discussion of the results is very short and general for such a large amount of experimental data. It needs to be expanded and detailed.
Author Response
The article is an interesting research, contains a large amount of data. However, the sequence of presentation and incomplete presentation of data complicates perception. So, for example, table 2 shows different indicators for male and female children, apparently in order to reduce the amount of data. In general, it seems to me that it would be more appropriate to use multidimensional methods of data processing in order to more clearly present information. The conclusions are too general from such a volume of information. I believe that the article needs revision in terms of presentation of the material and discussion of the results. There are typos in the article.
However, the authors of the article, in my opinion, chose an inappropriate way of presenting data, overloading the article with tables and figures.
1. Table 2 - only the differing values ​​for males and females are given, although it is interesting to understand the order of the differences. I would like to see the same list of parameters for both male and female.
2. Figure 1 - for children with VPI data are given for amino acids, for children with MLPI for acylcarnitines.
3. Table 3 - again a different list of parameters for each subgroup. I would recommend using a more visual way of presenting the data, for example, the PCA method. It will allow not only to reduce the dimension, i.e. reduce the number of important differing parameters, but also show their relationship.
We thank the Reviewer for the interesting suggestions. Tables and figures have been designed in this way in order to highlight only significant differences, avoiding long lists of not significant parameters. Furthermore, although the primary aim was represented by the assessment of differences in terms of levels of the parameters we collected, the suggestion of a multidimensional analysis is not supported by the study design and by technical limitations represented by the poor correlation between the described variables. To provide the reader with the complete picture of all the analyzed parameters, additional supplementary tables have been inserted showing the absolute values ​​of the metabolites and the relative P-value (Supplementary Tables 2,3,4,5,6,7,8,9). Conclusions have been deepened, as requested, and typos have been corrected.
- Very difficult to read table 5. How are p values ​​in the table calculated? Was the Bonferroni Amendment used? In this case, the Kruskal-Wallis test can be used, since the distribution of the data is most likely not normal. We thank the Reviewer for having raised this point. The comparison for quantitative and non-normal variables is based on Kruskal-Wallis, whereas for qualitative variables it is based on chi-squared test. The objective of the last part of the cluster analysis was to assess differences among clusters, considering the multiple comparisons.
- The discussion of the results is very short and general for such a large amount of experimental data. It needs to be expanded and detailed. Discussion has been improved, and some references included, but unfortunately, research on sex differences in infant metabolome is scarce, so that is not possible to discuss our findings in more depth based on the available literature.
Reviewer 3 Report
The paper show interesting data. The methodology look adequate, the statistical analysis is correct, but the discussion need to be improved in order to explain better the results. Some references that explain the metabolic profile must be included.
Author Response
The paper show interesting data. The methodology look adequate, the statistical analysis is correct, but the discussion need to be improved in order to explain better the results. Some references that explain the metabolic profile must be included. We thank the referee for understanding the meaning of our work and appreciating it. Discussion has been improved, and some references included, but unfortunately, research on sex differences in infant metabolome is scarce, so that is not possible to discuss our findings in more depth based on the available literature.
Reviewer 4 Report
The manuscript by Caterino et al entitled “Sex Affects Human Premature Neonates’ Blood Metabolome According to Gestational Age (GA), Parenteral Nutrition (PN), and Caffeine Treatment” aimed towards the assessment of the role played by GA in sex-related differences in the blood metabolome of premature neonates using a targeted metabolomic approach for the detection of amino acids and acylcarnitines in dried blood spots. Their study reports that the presence of a sex-dependent metabolome in premature infants, which is affected by GA and Caffeine. Although the authors have put in efforts in their study to define the sex-dependent metabolome of Human Premature Neonates through stratification of parameters such as GA, PN, and caffeine, the results obtain and the interpretation of their results rather seems incoherent. Moreover, the manuscript needs thorough proofreading and general formatting to be considered for publication.
Major comments
- There are many instances throughout the manuscript where authors have struggled with the English language, failing to structure and articulate proper meaningful sentences. Please consider revising and proofreading the manuscript from a native English speaker.
- Line 102, the authors report that VPI females had higher C5:1 when compared with MLPI. Refer to table 2 there is no change observed for C5:1. Please explain this discrepancy.
- Line 147-148, the authors report that, In female MLPI, PN increased C8DC (Table 3). Refer to table 3 there is no change observed for C8DC. Please explain this discrepancy
- Line 149-150, the authors report that, In male VPI, PN lowered C14-OH. Refer to table 3 there is no change observed for C14-OH. Please explain this discrepancy.
- Line 177-178 the authors report that Female VPI treated with caffeine resulted differently only for Glu and C5:1, whose concentrations were higher and lower when compared with those of non-treated individuals respectively. Refer to Figure3A the concentration values are lower for both Glu and C5:1. Please explain this discrepancy.
- Line 233-235, the result from their previous study [250] is included here which is kind of confusing since there is no continuity in this paragraph due to a lack of clear explanation of the results.
Minor comments
- The quality of images submitted both in manuscript and supplementary are not legible enough.
- Figure 2 Box plot for C8 needs more enlargement on the y-axis.
- There are several punctuation mistakes in supplemental figures eg. Supplemental figure 2E, C18:1OH, C6:1 is written as C181OH, C61. C18:2, C18:1, C14:1 is written as C182, C181, C141 in Supplementary Figure S3.
- PLS-DA is referred to as DA multiple times. Please maintain consistency while reporting.
- Line 192 Supplementary Figure Please correct the numbering.
Author Response
The manuscript by Caterino et al entitled “Sex Affects Human Premature Neonates’ Blood Metabolome According to Gestational Age (GA), Parenteral Nutrition (PN), and Caffeine Treatment” aimed towards the assessment of the role played by GA in sex-related differences in the blood metabolome of premature neonates using a targeted metabolomic approach for the detection of amino acids and acylcarnitines in dried blood spots. Their study reports that the presence of a sex-dependent metabolome in premature infants, which is affected by GA and Caffeine. Although the authors have put in efforts in their study to define the sex-dependent metabolome of Human Premature Neonates through stratification of parameters such as GA, PN, and caffeine, the results obtain and the interpretation of their results rather seems incoherent. Moreover, the manuscript needs thorough proofreading and general formatting to be considered for publication.
Major comments
- There are many instances throughout the manuscript where authors have struggled with the English language, failing to structure and articulate proper meaningful sentences. Please consider revising and proofreading the manuscript from a native English speaker. As requested, the text has been revised by a native English speaker.
- Line 102, the authors report that VPI females had higher C5:1 when compared with MLPI. Refer to table 2 there is no change observed for C5:1. Please explain this discrepancy; 3. Line 147-148, the authors report that, In female MLPI, PN increased C8DC (Table 3). Refer to table 3 there is no change observed for C8DC. Please explain this discrepancy; 4. Line 149-150, the authors report that, In male VPI, PN lowered C14-OH. Refer to table 3 there is no change observed for C14-OH. Please explain this discrepancy. We thank the referee for the observation, but there is no discrepancy: the statistical analysis is based on the comparison between the ranks and provides a significant P value based on IQR which shows who is higher or lower. Therefore, our interpretation is correct.
- Line 177-178 the authors report that Female VPI treated with caffeine resulted differently only for Glu and C5:1, whose concentrations were higher and lower when compared with those of non-treated individuals respectively. Refer to Figure3A the concentration values are lower for both Glu and C5:1. Please explain this discrepancy. We apologise for the oversight, concentrations were both lower, and the text has been corrected accordingly.
- Line 233-235, the result from their previous study [250] is included here which is kind of confusing since there is no continuity in this paragraph due to a lack of clear explanation of the results. To avoid confusion, the reference to the previous study has been deleted.
Minor comments
- The quality of images submitted both in manuscript and supplementary are not legible enough. The quality of images has been improved, and single file for each image has been provide separately.
- Figure 2 Box plot for C8 needs more enlargement on the y-axis. As requested, Box plot for C8 has been enlarged.
- There are several punctuation mistakes in supplemental figures eg. Supplemental figure 2E, C18:1OH, C6:1 is written as C181OH, C61. C18:2, C18:1, C14:1 is written as C182, C181, C141 in Supplementary Figure S3. We apologise for the oversight, supplementary figures have been now corrected.
- PLS-DA is referred to as DA multiple times. Please maintain consistency while reporting. Da has been replaced by PLS-DA throughout the text
- Line 192 Supplementary Figure Please correct the numbering. We apologise for the oversight, the figure number has been now corrected.
Round 2
Reviewer 2 Report
The authors made significant changes to the article in accordance with the comments of the reviewers. I believe that in its present form it is possible to recommend accepting the article for publication.
Reviewer 4 Report
The revised manuscript has addressed multiple points raised in initial review and is vastly improved. But there are still few instances in the manuscript having grammatical errors and typos. Also, in supplementary figure S3D the dotted circle is missing C4DC point.